# A Mechanism for Solving Relational Tasks in Transformer Language Models

## Abstract

A primary criticism towards language models (LMs) is their inscrutability. This paper presents evidence that, despite their size and complexity, LMs sometimes exploit a simple computational mechanism to solve one-to-one relational tasks (e.g., capital_of(Poland)=Warsaw). We investigate a range of language model sizes (from 124M parameters to 176B parameters) in an in-context learning setting, and find that for a variety of tasks (involving capital cities, upper-casing, and past-tensing) a key part of the mechanism reduces to a simple linear update typically applied by the feedforward (FFN) networks. These updates also tend to promote the output of the relation in a content-independent way (e.g., encoding Poland:Warsaw::China:Beijing), revealing a predictable pattern that these models take in solving these tasks. We further show that this mechanism is specific to tasks that require retrieval from pretraining memory, rather than retrieval from local context. Our results contribute to a growing body of work on the mechanistic interpretability of LLMs, and offer reason to be optimistic that, despite the massive and non-linear nature of the models, the strategies they ultimately use to solve tasks can sometimes reduce to familiar and even intuitive algorithms.

## 1 Intro

The growing capabilities of large language models (LLMs) have led to an equally growing interest in understanding how such models work under the hood. Such understanding is critical for ensuring that LLMs are reliable and trustworthy once deployed. Recent work (often now referred to as "mechanistic interpretability") has contributed to this understanding by reverse-engineering the data structures and algorithms that are implicitly encoded in the model's weights, e.g., by identifying detailed circuits (Wang et al., 2022; Elhage et al., 2021; Olsson et al., 2022) or by identifying mechanisms for factual storage and retrieval which support intervention and editing (Geva et al., 2021b; Li et al., 2022; Meng et al., 2022a;c; Dai et al., 2022).

Here, we contribute to this growing body of work by analyzing how LLMs recall information during in-context learning. Modern LLMs are based on a complex transformer architecture (Vaswani et al., 2017) which produces contextualized word embeddings (Peters et al., 2018; Devlin et al., 2019) connected via multiple non-linearities. Despite this, we find that LLMs implement a basic vector-addition mechanism which plays an important role in a number of in-context-learning tasks.

We study this phenomenon in three tasks: recalling capital cities, uppercasing tokens, and past-tensing verbs. Our key findings are:

- We find evidence of a distinct processing signature in the forward pass which characterizes this mechanism. That is, if models need to perform the get_capital($x$) function, which takes an argument $x$ and yields an answer $y$, they first surface the argument $x$ in earlier layers which enables them to apply the function and yield $y$ as the final output (Figure 2). This signature generalizes across models and tasks, but appears to become sharper as models increase in size.

- We take a closer look at GPT2-Medium, and find that the vector arithmetic mechanism is often implemented by mid-to-late layer feedforward networks (FFNs) in a way that is modular and supports intervention. E.g., an FFN outputs a content-independent update which produces Warsaw given Poland and can be patched into an unrelated context to produce Beijing given China.

- We demonstrate that this mechanism is specific to recalling information from pre-training memory. For settings in which the correct answer can be retrieved from the prompt, this mechanism does not appear to play any role, and FFNs can be ablated entirely with relatively minimal performance degradation. Thus, we present new evidence supporting the claim that FFNs and attention specialize for different roles, with FFNs supporting factual recall and attention copying and pasting from local context.

Taken together, our results offer new insights about one component of the complex algorithms that underlie in-context learning. While the mechanism we present occupies the narrow range of one-to-one relations, the simplicity of the mechanism raises the possibility that other apparently complicated behaviors may be supported by a sequence of simple operations under the hood. Moreover, our results suggest a distinct processing signature and hint at a method for intervention. These ideas could support future work on detecting and preventing unwanted behavior by LLMs at runtime.

## 2 Methods

In decoder-only transformer language models (Vaswani et al., 2017), a sentence is processed one word at a time, from left to right. In this paper, we focus on the transformations that the next-token prediction undergoes in order to predict the answer to some task. At each layer, an attention module and feed-forward network (FFN) module apply subsequent updates to this representation. Consider the FFN update at layer $i$, where $x_i$ is the current next-token representation. The update applied by the FFN here is calculated as $\text{FFN}(\vec{x_i}) = \vec{o_i}, \quad \vec{x_{i+1}} = \vec{x_i} + \vec{o_i}$ where $x_{i+1}$ is the updated token for the next layer. Due to the residual connection, the output vector $\vec{o_i}$ is added to the input. $\vec{x}$ is updated this way by the attention and FFNs until the end of the model, where the token is decoded into the vocab space with the language modeling head $E$: $\text{softmax}(E\vec{x})$. From start to end, $x$ is only updated by additive updates, forming a residual stream (Elhage et al., 2021). Thus, the token representation $x_i$ represents all of the additions made into the residual stream up to layer $i$.

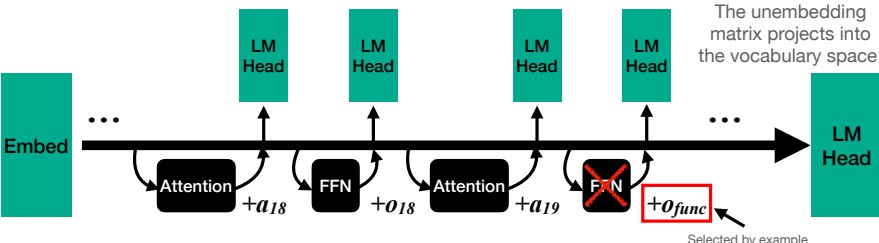

Figure 1: When decoding the next word, additive updates are made through the residual connections of each attention/FFN sub-layer. To decode the running prediction at every layer, the pre-trained language modeling head is applied at various points in each layer as in Geva et al. (2022a); nostalgebraist (2020). The $\vec{o}$ vector interventions we make (§4.1) are illustrated by patching one or more FFN outputs with one from another example

### 2.1 Early Decoding

A key insight from the residual stream perspective is that we can decode the next token prediction with the LM head before it reaches the final layer. This effectively allows for "print statements" throughout the model's processing. The intuition behind this technique is

that LMs incrementally update the token representation $\vec{x}$ to build and refine an encoding of the vocabulary distribution. This technique was initially introduced in nostalgebraist (2020) as the logit lens, and Geva et al. (2022b) show that LMs do in fact refine the output distribution over the course of the model. Figure 1 illustrates the process we use to decode hidden states into the vocabulary space using the pre-trained language modeling head $E$. After decoding, we apply a softmax to get a probability distribution over all tokens. When we decode at some layer, we say that the most likely token in the resulting vocab distribution is currently being represented in the residual stream. We examine the evolution of these predictions over the course of the forward pass for several tasks.

## 2.2 Tasks

We apply early decoding to suite of in-context learning tasks to explore the transformations the next token prediction undergoes in order to predict the answer.

World Capitals The World Capitals task requires the model to retrieve the capital city for various states and countries in a few-shot setting. The dataset we use contains 248 countries and territories. A one-shot example is shown below:

| |
|---|
| "Q: What is the capital of France? A: Paris Q: What is the capital of Poland? A:＿＿" Expected Answer: " Warsaw" |

Reasoning about Colored Objects We focus on a subset of 200 of the reasoning about colored objects dataset prompts (i.e., the colored objects dataset) from BIG-Bench (Srivastava et al., 2022). A list of colored common objects is given to the model before being asked about one object's color. For the purposes of this paper, we focus only on one aspect of this task–the model's ability to output the final answer in the correct format.[1]

| |
|---|
| "Q: On the floor, I see a silver keychain, a red pair of sunglasses, a gold sheet of paper, a black dog leash, and a blue cat toy. What color is the keychain? 
 A: Silver 
 Q: On the table, you see a brown sheet of paper, a red fidget spinner, a blue pair of sunglasses, a teal dog leash, and a gold cup. What color is the sheet of paper? 
 A:＿＿" Expected answer: " Brown" |

The above tasks could all be described as one-to-one (e.g., each country has one capital, each word only has one uppercase form). In Appendix F we explore six additional tasks, three of which are either many-to-many or many-to-one. We find that the observed mechanism only applies to one-to-one relations, indicating that the model learns some sensitivity to this type of relation in order for it to represent the structure required for the mechanism described here. Past Tense Verb Mapping Lastly, we examine whether an LM can accurately predict the past tense form of a verb given a pattern of its present tense. The dataset used is the combination of the regular and irregular partitions of the past tense linguistic mapping task in BIG-Bench (Srivastava et al., 2022). After filtering verbs in which the present and past tense forms start with the same token, we have a total of 1,567 verbs. An example one-shot example is given below:

| |
|---|
| "Today I abandon. Yesterday I abandoned. Today I abolish. Yesterday I＿＿" Expected answer: " abolished" |

## 2.3 Models

We experiment exclusively on decoder-only transformer LMs across various sizes and pre-training corpora. When not specified, results in figures are from GPT2-medium. We also include results portraying the stages of processing signatures in the residual streams of the small, large, and extra large variants (Radford et al.), the 6B parameter GPT-J model (Wang & Komatsuzaki, 2021), and the 176B BLOOM model (Scao et al., 2022), either in the main paper or in the Appendix.

---

[1]The reason for this is that most of the results in this paper were originally observed as incidental findings while studying the Reasoning about Colored Objects task more generally. We thus zoom in on this one component for the purposes of the mechanism studied here, acknowledging that the full task involves many other steps that will no doubt involve other types of mechanisms.

# 3 Stages of Processing in Predicting the Next Token

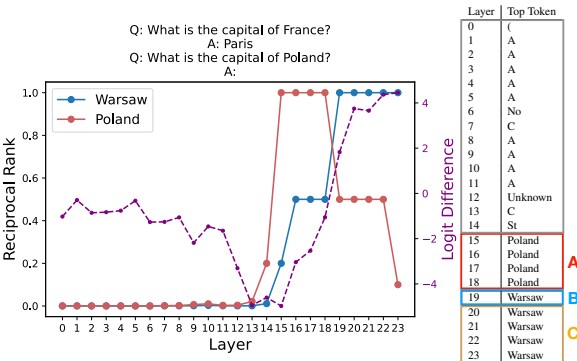

Figure 2: Decoding the next token prediction at each layer reveals distinct stages of processing. The blue box (A) shows where the model prepares an argument for transformation, the red box (B) shows the function application phase during which the argument is transformed (here with the capital_of function, and the yellow box (C) shows a saturation event, in which the model has found the answer, and stops updating the top prediction. The dashed line shows the logit difference between argument and answer at each layer.

First, we use the early decoding method in order to investigate how the processing proceeds over the course of a forward pass to the model. Each task requires the model to infer some relation to recall some fact, e.g., retrieving the capital of Poland. In these experiments, we see several discrete stages of processing that the next token undergoes before reaching the final answer. These states together provide evidence that the models "apply" the relevant functions (e.g., get_capital) abruptly at some mid-late layer to retrieve the answer. Moreover, in these cases, the model prepares the argument to this function in the layers prior to that in which the function is applied.

In Figure 2 we illustrate an example of the stages we observe across models. For the first several layers, we see no movement on the words of interest. Then, during Argument Formation, the model first represents the argument to the desired relation in the residual stream. This means that the top token in the vocabulary distribution at some intermediate layer(s) is the subject the question inquires about (e.g., the $x$, in get_capital($x$). During Function Application we find that the model abruptly switches from the argument to the output of the function (the $y$, in get_capital($x$) = $y$). We find that function application is typically applied by the FFN update at that layer to the residual stream. This is done by adding the output vector $\vec{o}$ of the FFN to the residual stream representation, thus transforming it with an additive update. We study these $\vec{o}$ vectors in detail in Section 4. Finally, the model enters Saturation[2], where the model recognizes it has solved the next token, and ceases updating the token representation for the remaining layers.

The trend can be characterized by an X-shaped pattern of the argument and final output tokens when plotting the ranks of the argument($x$) and output ($y$) tokens. We refer to this behavior as argument-function processing. Figure 3 shows that this same processing signature can be observed consistently across tasks and models. Moreover, it appears to become more prominent as the models increase in size. Interestingly, despite large differences in number of layers and overall size, models tend to undergo this process at similar points proportionally in the model.

# 4 Implementation of Context-Independent Functions in FFN Updates

The above results on processing signature suggest that the models "apply" a function about 2/3rds of the way through the network with the addition of an FFN update. Here, we investigate the mechanism via which that function is applied more closely. Specifically, focusing on GPT2-Medium[3], we show that we can force the encoded function to be applied to new arguments in new contexts by isolating the responsible FFN output vector and then dropping into a forward pass on a new input.

---

[2]Saturation events are described in Geva et al. (2022a) where detection of such events is used to "early-exit" out of the forward pass

[3]We focus on one model because manual analysis was required in order to determine how to perform the intervention. See Appendix for results on GPT-J and Section 7 for discussion.

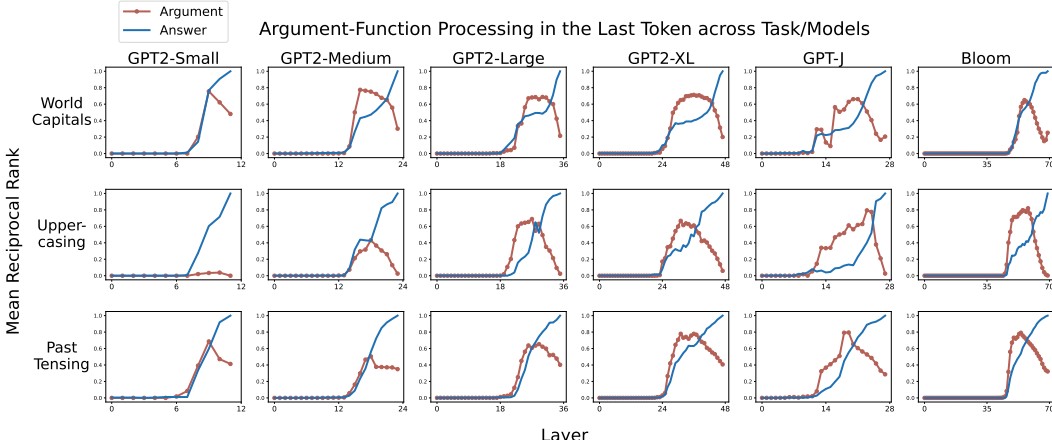

Figure 3: Argument formation and function application is characterized by a promotion of the argument (red) followed by it being replaced with the answer token (blue), forming an X when plotting reciprocal ranks. Across the three tasks we evaluate, we see that most of the models exhibit these traces, and despite the major differences in model depths, the stages occur at similar points in the models. Data shown is filtered by examples in which the models got the correct answer.

## 4.1 $\vec{o}$ Vector Interventions

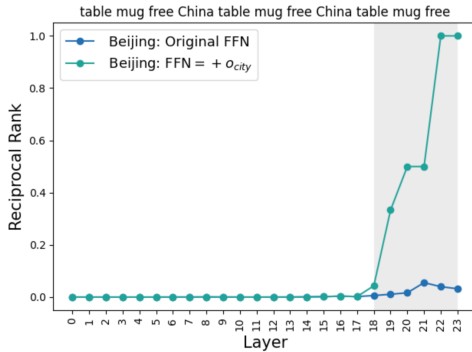

Figure 4: The gray area indicates layers with the FFN intervention. Even if the input context is nonsense (repeating pattern), when "China" is represented in the residual stream, the $o_{city}$ vector promotes the correct capital city.

Consider the example in Figure 2. At layer 18, the residual stream ($\vec{x_{18}}$) is in argument formation, and represents the " Poland" token. At the end of layer 19, a function is applied, transforming $\vec{x_{19}}$ into the answer token " Warsaw.

As discussed in the previous section, we can isolate the function application in this case to FFN 19; let $\tilde{x_{19}}$ represent the residual stream after the attention update, but before the FFN update at layer 19 (which still represents Poland). Recall that the update made by FFN 19 is written $\mathrm{FFN}_{19}(\tilde{x_{19}}) = \vec{o_{19}}$ and $\vec{x_{19}} = \tilde{x_{19}} + \vec{o_{19}}$. We find that $\vec{o_{19}}$ will apply the get_capital function regardless of the content of $\tilde{x_{19}}$. For example, if we add $\vec{o_{19}}$ to some $\tilde{x}$ which represents the " China" token, it will transform into " Beijing". Thus we refer to $\vec{o_{19}}$ as $\vec{o_{city}}$ since it retrieves the capital cities of locations stored in the residual stream. We locate such $\vec{o}$ vectors in the uppercasing and past tense mapping tasks in the examples given in Section 2.2, which we refer to as $\vec{o_{upper}}$ and $\vec{o_{past}}$, respectively.[4]

We test whether these updates have the same effect, and thus implement the same function, as they do in the original contexts from which they were extracted. To do so, we replace entire FFN layers with these vectors and run new inputs through the intervened model.[5]

---

[4]In Appendix A, we extend these results to GPT-J, for which the same procedure leads to strong effects on uppercasing, but smaller overall positive effects on capital cities and past tensing (see Section 7).

[5]Which FFNs to replace is a hyperparameter; we find that replacing layers 18-23 in GPT2-Medium leads to good results. It also appears necessary to replace multiple FFNs at a time. See additional experiments in Appendix D. In summary, it is likely that the $\vec{o}$ vectors are added over the course

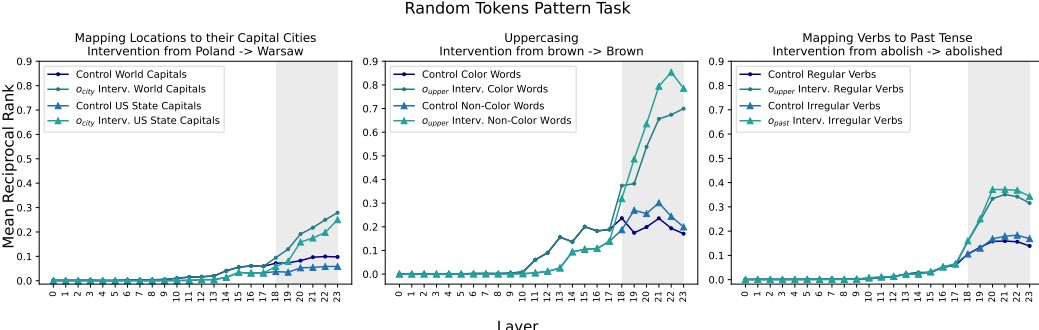

Figure 5: We intervene on GPT2-Medium's forward pass while it is predicting the completion of a pattern. The control indicates normal model execution, while the gray boxes indicate which FFNs are replaced with our selected $\vec{o}$ vectors. We can see a significant increase in the reciprocal rank of the output of the function implemented by the $\vec{o}$ vector used even though the context is completely absent of any indication of the original task.

Data: We are interested in whether the captured o vectors can be applied in a novel context, in particular, to a context that is otherwise devoid of cues as to the function of interest. Thus, we synthesize a new dataset where each entry is a string of three random tokens (with leading spaces) followed by a token $x$ which represents a potential argument to the function of interest. For example, in experiments involving $o_{city}$, we might include a sequence such as table mug free China table mug free China table mug free. This input primes the model to produce "China" at the top of the residual stream, but provides no cues that the capital city is relevant, and thus allows us to isolate the effect of $o_{city}$ in promoting "Beijing" in the residual stream. In addition to the original categories, we also include an "out-of-domain" dataset for each task: US states and capitals, 100 non-color words, and 128 irregular verbs. These additional data test the sensitivity of the $\vec{o}$ vectors to different types of arguments.

Results: Figure 4 shows results for a single example. Here, we see that "Beijing" is promoted all the way to the top of the distribution solely due to the injection of $\vec{o_{city}}$ into the forward pass. Figure 5 shows that this pattern holds in aggregate. In all settings, we see that the outputs of the intended functions are strongly promoted by adding the corresponding $\vec{o}$ vectors. By the last layer, for world and state capitals, the mean reciprocal rank of the target city name across all examples improves from roughly the 10th to the 3rd-highest ranked word and 17th and 4th-ranked words respectively. The target output token becomes the top token in 21.3%, 53.5%, and 7.8% of the time in the last layer in the world capitals, uppercasing, and past tensing tasks, respectively.

We also see the promotion of the proper past tense verbs by $\vec{o_{past}}$. The reciprocal ranks improve similarly for both regular (approx. 7th to 3rd rank) and irregular verbs (approx. 6th to 3rd), indicating that the relationship between tenses is encoded similarly by the model for these two types. $\vec{o_{upper}}$ promotes the capitalized version of the test token almost every time, although the target word starts at a higher rank (on average, rank 5). These results together show that regardless of the surrounding context, and regardless of the argument to which it is applied, $\vec{o}$ vectors consistently apply the expected functions. Since each vector was originally extracted from the model's processing of a single naturalistic input, this generalizability suggests cross-context abstraction within the learned embedding space.

Common Errors: While the above trend clearly holds on the aggregate, the intervention is not perfect for individual cases. The most common error is that the intervention has no real effect. In the in-domain (out-domain) settings, this occurred in about 37% (20%) of capital cities, 4% (5%) on uppercasing, and 19% (22%) for past tensing. We believe the rate is so much higher for world capitals because the model did not have a strong association between certain country-capital pairs from pretraining, e.g, for less frequently mentioned countries.

of several layers, consistent with the idea that residual connections encourage each layer to move gradually towards a point of lower loss (Jastrzebski et al., 2017).

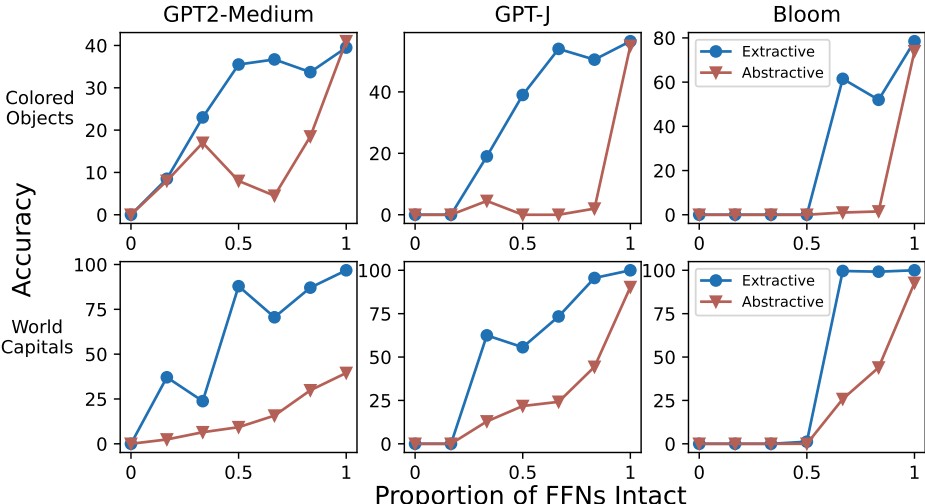

Figure 6: Removing FFNs negatively affects performance when the task is abstractive: the in-context label is an out-of-context transformation of the in-context prompt (e.g., " silver" in context, answer given as " Silver"). In comparison, on the extractive dataset, performance is robust to a large proportion of FFNs being removed. Other models tested are shown in Appendix B

Typically, in these cases, the top token remains the argument, but sometimes becomes some random other city, for example, predicting the capital of Armenia is Vienna. We also find that the way tokenization splits the argument and target words affects the ability of the $\vec{o}$ vector to work and is another source of errors. This is discussed further in Appendix E.

## 5 The Role of FFNs in Out-of-Context Retrieval

So far, we have shown that FFN output vectors can encode functions that transfer across contexts. Here, we investigate the role of this mechanism when we control whether the amswer occurs in context. The tasks we study previously require recalling a token that does not appear in the given context (abstractive tasks). In this section we show that mid-higher layer FFNs are crucial for this process. When the answer to the question does appear in context (extractive tasks), we find that ablating a subset of FFNs has a comparatively minor effect on performance, indicating that they are relatively modular and there is a learned division of labor within the model. This observation holds across the decoder-only LMs tested in this paper. This breakdown is consistent with previous work finding that FFNs store facts learned from pre-training (Geva et al., 2021a; Meng et al., 2022b;c) and attention heads copy from the previous context (Wang et al.; Olsson et al., 2022).

### 5.1 Abstractive vs. Extractive Tasks

Extractive Tasks: Extractive tasks are those in which the exact tokens required to answer a prompt can be found in the input context. These tasks can thus be solved by parsing the local context alone, and thus do not necessarily require the model to apply a function of the type we have focused on in this paper (e.g., a function like get_capital).

Abstractive Tasks: Are those in which the answer to a prompt is not given in the input context and must be retrieved from pretraining memory. Our results suggest this is done primarily through argument-function processing, requiring function application through (typically) FFN updates as described in Section 3.

We provide examples with their associated GPT2-Medium layerwise decodings in Figure 7. We expect that the argument formation and function application stages of processing occur primarily in abstractive tasks. Indeed, in Appendix A, we show that the characteristic argument-answer X pattern disappears on extractive inputs. We hypothesize that applying out-of-context transformations to the predicted token representation is one of the primary functions of FFNs in the mid-to-late layers, and that removing them should only have a major effect on tasks that require out-of-context retrieval.

## 5.2 Effect of Ablating FFNs

Data: Consider the example shown in Section 2.2 demonstrating the $\vec{o_{upper}}$ function. By providing the answer to the in-context example as " Silver", the task is abstractive by requiring the in-context token " brown" to be transformed to " Brown" in the test example. However, if we provide the in-context label as " silver", the task becomes extractive, as the expected answer becomes " brown". We create an extractive version of this dataset by lowercasing the example answer. All data is presented to the model with a single example (one-shot). The abstractive and extractive examples differ by only this single character.

| Layer | Top Tokens per Layer | |
| | Abstractive Task | Extractive Task |
| --- | --- | --- |
| | Q: What is the capital of Somalia? A: Mogadishu Q: What is the capital of Poland? A: | The capital of Somalia is Mogadishu. The capital of Poland is Warsaw. Q: What is the capital of Somalia? A: Mogadishu Q: What is the capital of Poland? A: |
| ... | ... | ... |
| 14 | St | St |
| 15 | Poland | St |
| 16 | Poland | Warsaw |
| 17 | Poland | Warsaw |
| 18 | Poland | Warsaw |
| 19 | Warsaw | Warsaw |
| 20 | Warsaw | Warsaw |
| 21 | Warsaw | Warsaw |
| 22 | Warsaw | Warsaw |
| 23 | Warsaw | Warsaw |

Figure 7: The abstractive task undergoes argument formation and function application, while the extractive task immediately saturates (yellow).

We repeat this experiment on the world capitals (see Figure 7). Notice, however, that since the answer is provided explicitly, the task is much easier for the models in the extractive case.

Results: We run the one-shot extractive and abstractive datasets on the full models, and then repeatedly remove an additional 1/6th of all FFNs from the top down (e.g., in 24 layer GPT2-Medium: removing the 20-24th FFNs, then the 15-24th, etc.). Our results are shown in Figure 6. Despite the fact that the inputs in the abstractive and extractive datasets only slightly differ (by a single character in the colored objects case) we find that performance plummets on the abstractive task as FFNs are ablated, while accuracy on the extractive task drops much more slowly. For example, even after 24 FFN sublayers are removed from Bloom (totaling ~39B parameters) extractive task accuracy for the colored objects dataset drops 17% from the full model's performance, while abstractive accuracy drops 73% (down to 1% accuracy). The case is similar across model sizes and pretraining corpora; we include results on additional models in Appendix B. This indicates that we can isolate the effect of locating and retrieving out of context tokens in this setting to the FFNs. Additionally, because the model retains reasonably strong performance compared to using the full model, we do not find convincing evidence that the later layer FFNs are contributing to the extractive task performance, supporting the idea of modularity within the network.

## 6 Related Work

Attributing roles to components in pretrained LMs is a widely studied topic. In particular, the attention layers (Olsson et al., 2022; Kobayashi et al., 2020; Wang et al.) and most relevant to this work, the FFN modules, which are frequently associated with factual recall and knowledge storage (Geva et al., 2021a; Meng et al., 2022a;c). How language models store and use knowledge has been studied more generally as well (Petroni et al., 2019; Cao et al., 2021; Dai et al., 2022; Bouraoui et al., 2019; Burns et al., 2022; Dalvi et al., 2022; Da et al., 2021) as well as in static embeddings (Dufter et al., 2021). Recent work in mechanistic interpretability aims to fully reverse engineer how LMs perform some behaviors (Elhage et al., 2021). Our work builds on the finding that FFN layers promote concepts in the vocabulary space (Geva et al., 2022a) by breaking down the process the model uses to do this in context;

Bansal et al. (2022) perform ablation studies to test the importance of attention and FFN layers on in-context learning tasks. Other work analyze information flow within an LM to study how representations are built through the layers, finding discrete processing stages (Voita et al., 2019; Tenney et al., 2019). We also follow this approach, but our analysis focuses on interpreting how models use individual updates within the forward pass, rather than probing for information encoded within some representation. Ilharco et al. (2023) show that vector arithmetic can be performed with the weights of finetuned models to compose tasks, similar to how $\vec{o}$ vectors can induce functions in the activation space of the model.

## 7    Discussion

In this work, we describe a mechanism that appears to help LMs recall relational information by transforming the next-token prediction through several discrete stages of processing. A core challenge in interpreting neural networks is determining whether the information attributed to certain model components is actually used for that purpose during inference (Hase & Bansal, 2022; Leavitt & Morcos, 2020). While previous work has implicated FFNs in recalling factual associations (Geva et al., 2022a; Meng et al., 2022a), we show through intervention experiments that we can manipulate the information flowing through the model according to these stages. This process provides a simple explanation for the internal subprocesses used by LMs on one type of problem. Our findings invite future work aimed at understanding why, and under what conditions, LMs learn to use this mechanism when they are capable of solving such tasks using, e.g., adhoc memorization.

A limitation that we observe is that the process for carrying out the $\vec{o}$ intervention depends on hyperparameters which are often model-specific (i.e., the exact stimuli used to extract the intervention, and the layer(s) at which to perform the intervention). We provide our most detailed investigation on GPT2-Medium, which clearly illustrates the phenomenon. Our experiments on stages of processing with GPT-J suggest that the same phenomena is in play, although (as discussed in Section 4 and Appendix A), the procedures we derive for interventions on GPT2-Medium do not transfer perfectly. Specifically, we can strongly reproduce the intervention results on uppercasing for GPT-J; results on the other two tasks are positive but with overall weaker effects. This requirement of model-specific customization is common in similar mechanistic interpretability work, e.g., (Wang et al., 2022; Geva et al., 2022b), and a priority in future work must be to generalize specific findings to model-agnostic phenomena. That said, in this work and other similar efforts, a single positive example as a proof of concept is often sufficient to advance understanding and spur future work that improves robustness across models.

In the long term, if we can understand how models break down complex problems into simple and predictable subprocesses, we can help more readily audit their behavior. Interpreting the processing signatures of model behaviors might offer an avenue via which to audit and intervene at runtime in order to prevent unwanted behavior. Moreover, understanding which relations FFNs encode could aid work in fact location and editing. Contemporaneous work (Geva et al., 2023) has studied a different mechanism for factual recall in LMs, but it is unclear how and when these mechanisms interact.

## 8    Conclusion

We contribute to a growing body of work on interpreting how the internal processes of language models (LMs) produce some behavior. On three in-context learning tasks, we observe that the next-token prediction appears to undergo several stages of processing in which LMs represent arguments to functions in their residual streams. This process occurs in models ranging in size from 124M to 176B parameters. On GPT2, We study instances where the additive update is made by the output vectors ($\vec{o}$ vectors) of feed-forward networks (FFNs). We show that for all tasks we test, $\vec{o}$ vectors calculated by the model in the process of solving some task can be extracted and replace the FFN updates of the model to solve novel instances of that task, providing evidence that LMs can learn self-contained and context-independent functions from pretraining.

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

## A    Argument-Function Processing in Other Models

In Section 3 we show that GPT2-Medium and Bloom promote the in-context 'argument' token to some function before promoting the answer to that function. In figure 8we show that this effect is present across other models as well in the three tasks we test. Qualitatively, we find that the pattern is more prominent in models that have more layers, likely because we are able to get more measurements after the FFN updates, so it is less likely that entire argument formation stage happens within a single layer (i.e., after the attention module update – we only take measurements after the FFN update for simplicity). In the extractive task setting, we would not expect the model to go through argument-function processing in order to reach the prediction, since it already appears in context (although this does not preclude it from doing so – it is still a valid way to retrieve the required information). We see that this X shaped pattern disappears when we plot the argument-answer curves for the extractive world capitals data, as shown next to the abstractive setting in Figure 9.

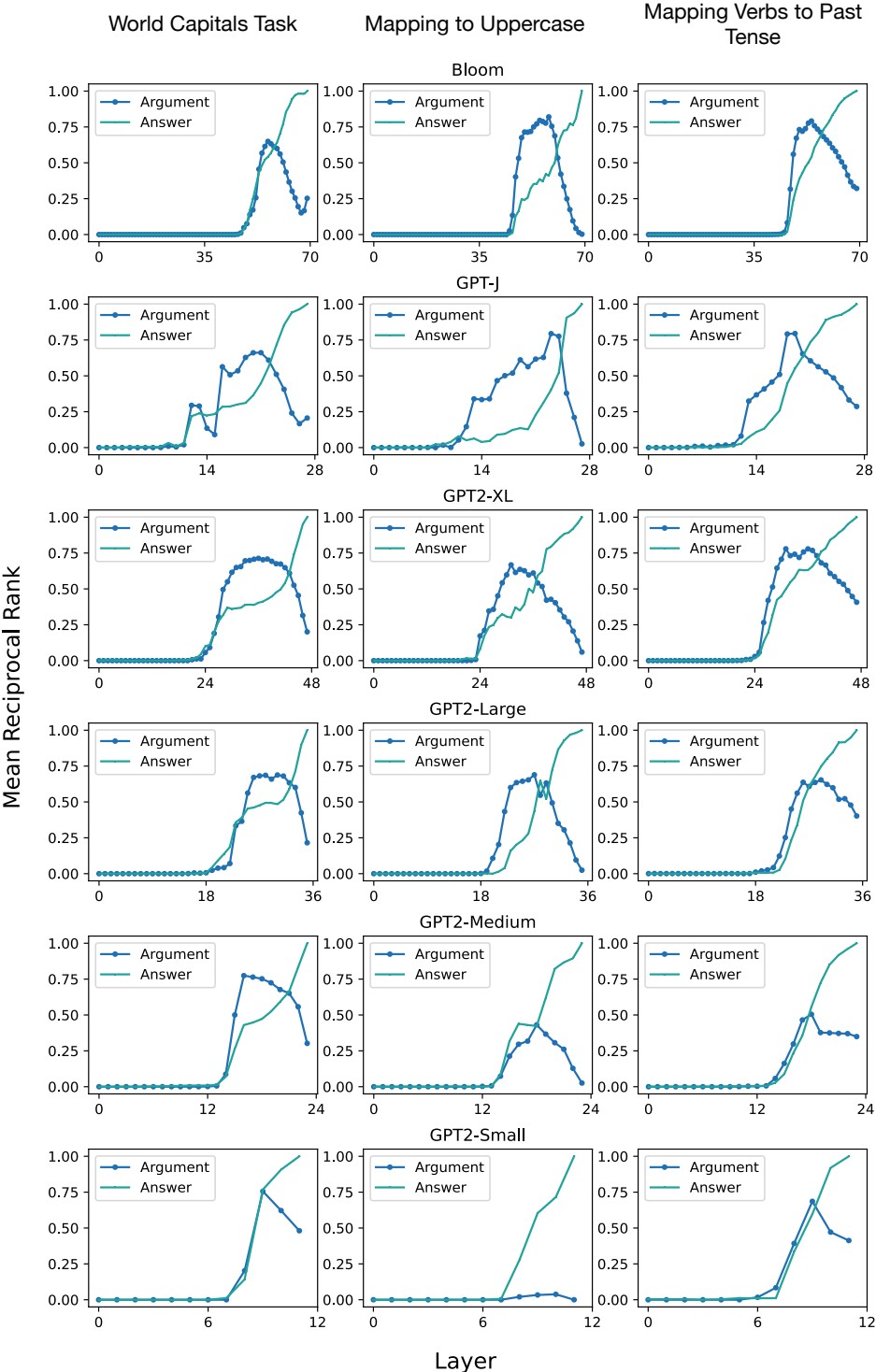

Figure 8: Across several model architectures and tasks, we find evidence that on average, the argument (which appears in context) rises to the top of the vocab distribution before crossing with the answer to the task. We describe this as argument-function processing where the argument to some function is represented in the residual stream before some update from the model is added to it to produce the output of that function. Qualitatively, we observe that models with more layers display this pattern more prominently.

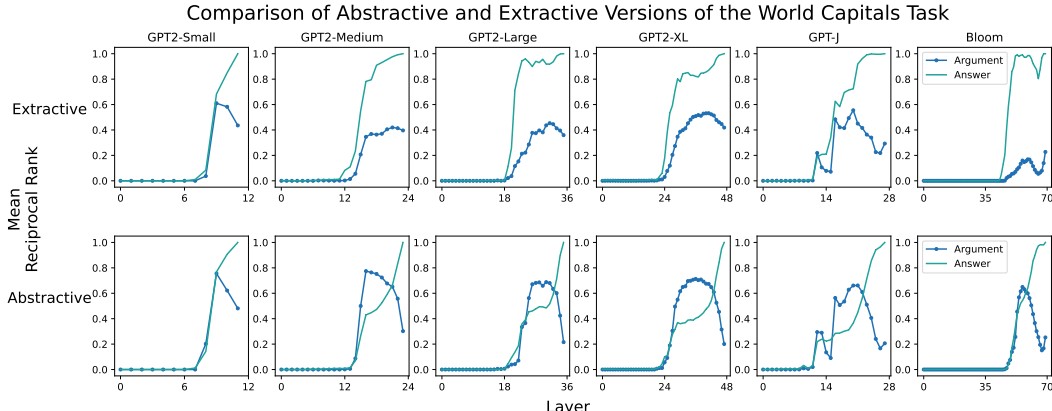

Figure 9: The 'X' pattern of argument and answer tokens crossing in the course of the forward pass is the characteristic pattern in argument-function processing. In the main text, we show how the models we test use this type of processing to recall the capital cities of locations. When we make the task extractive (by including the correct capital in the given context), the model does not have to setup an argument and function in order to get the answer, and the pattern disappears. This highlights the differences we describe in processing extractive and abstractive tasks. Both datasets are filtered for examples where the models were correct.

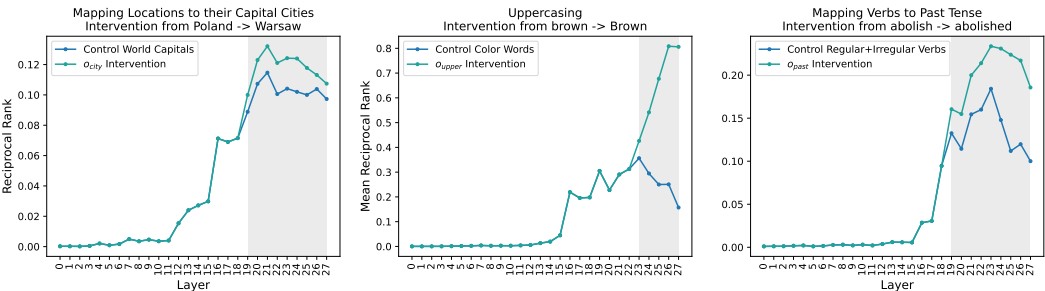

Figure 10: We use the same stimuli to extract $\vec{o}$ vectors on GPT-J. Results are similar for the uppercasing function, but only very weakly positive on the world capitals task.

We repeat the random tokens task on GPT-J using the same stimuli as in the main paper to select $\vec{o}$ vectors. We find that we can locate $\vec{o}$ vectors occurring in other models, however the success rate varies for the tasks that we evaluate in this work. Results are shown in Figure 10. Although the uppercasing function works very well, we get weaker responses for the past tense and world capitals mappings. One explanation could be that these tasks are not solved with an as-general solution as in GPT2, but the process for carrying out this intervention depends on hyperparameters which are often model-specific (i.e., the exact layer at which to perform the intervention), so future work is needed to understand where differences between these models lie.

## B  Additional Results on Ablating FFNs

We include the results for all six models we test for the FFN ablation study for both the colored objects task (Figure 11) and the world capitals task (Figure 12). We find that the trend of abstractive performance dropping off far before extractive performance is reflected across all models.

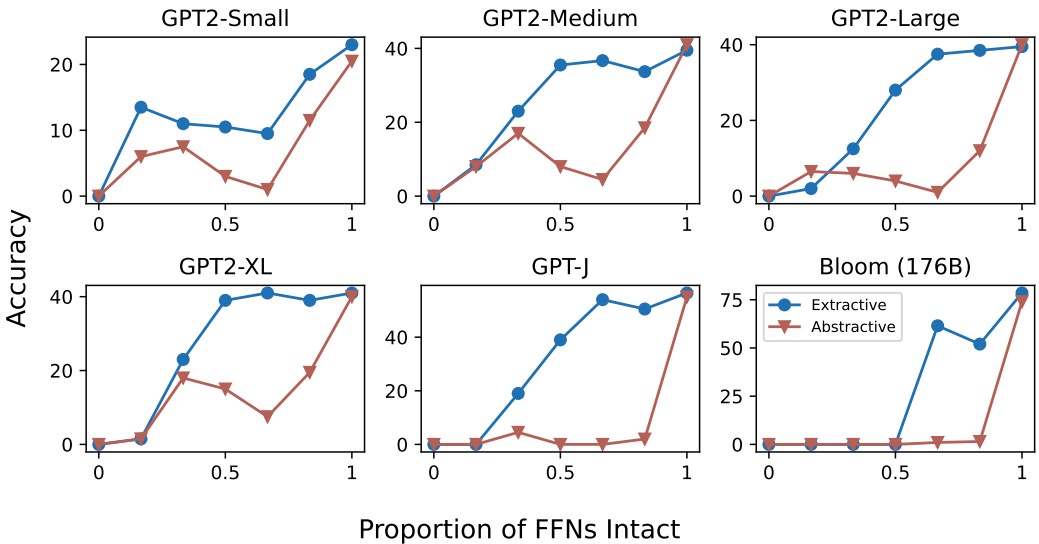

Figure 11: Results of removing FFN sublayers for the colored objects task for all models.

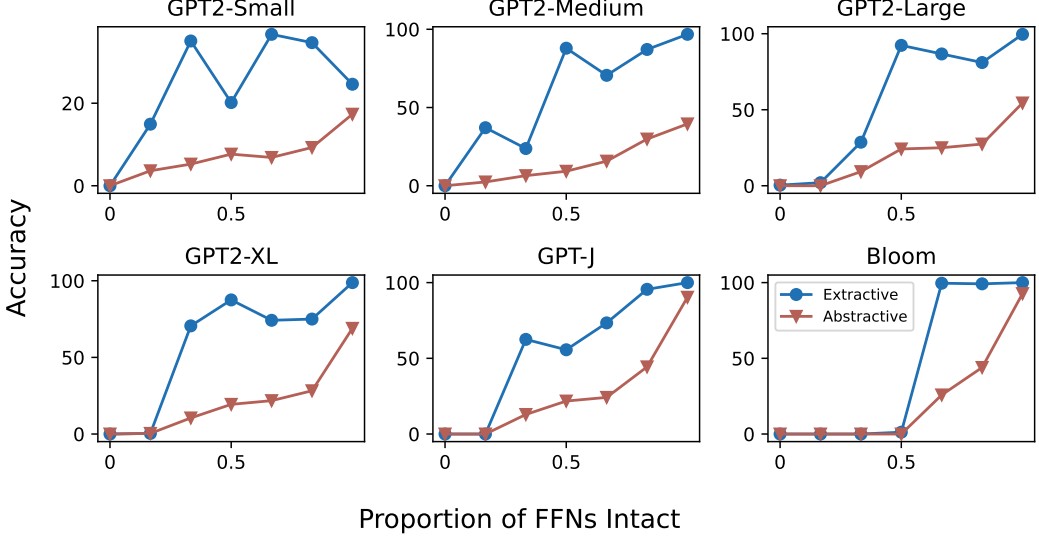

Figure 12: Results of removing FFN sublayers for the world capitals task for all models.

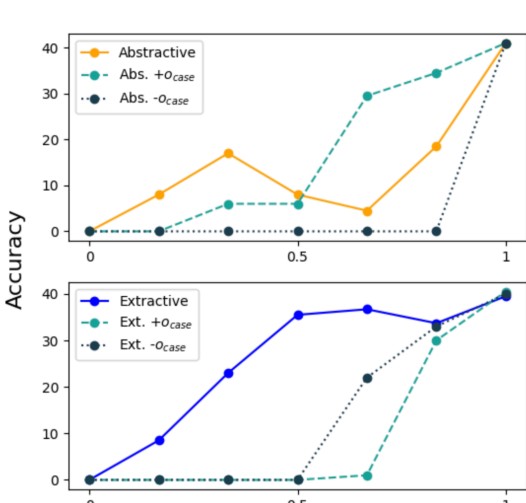

Figure 13: Replacing FFN updates with $+o_{case}$ helps recover accuracy in abstractive tasks where the answer is expected to be uppercase compared to subtracting it or ablating the FFNs. In extractive tasks, the task is primarily solved by attention modules and adding or subtracting $o_{case}$ only hurts performance.

### B.1   $+/-o_{case}$ Intervention on Colors

As illustrated in the example in Figure ??, adding $o_{case}$ to the residual stream ($x_{19} + o_{case}$) has the effect of capitalizing the first letter in the word 'brown'. Similar to the results in Sections 2.2 and 2.2, we find that adding $o_{case}$ to the residual stream has the effect of uppercasing the token prediction on arbitrary contextualized representations in the mid layers of GPT2-Medium. However, we also find that lowercasing the first letter can be accomplished by subtracting it. Qualitatively, this works much the same way as adding the $\vec{o}$ vectors previously discussed. We show this effect empirically, by showing the difference between replacing the FFN updates in GPT2-Medium with either positive or negative $o_ccase$ (having the effect of adding or subtracting from the residual stream).

We progressively remove FFNs from the top of the model, and show the effect of adding or subtracting $o_{case}$ in Figure 13. In the abstractive case, we find that accuracy is greatly boosted when adding $o_{case}$ which we identify as implementing an uppercasing function, and reflects the results in Sections 2.2 and 2.2. We find that we can replace the top third of GPT2-Medium FFN layers (FFNs in layers 16-24, around 20% of all parameters) with $+o_{case}$ to gain 25% in total accuracy (from 4.5% to 29.5%) and recovering to 72% of the performance of the un-ablated model (41%). Conversely, if we subtract $o_{case}$ in the abstractive setting to encourage lowercasing (i.e., encouraging the model to output a lowercased answer when the answer it should have a capital first letter), the model immediately hits 0% performance. We see the opposite effect in the extractive setting, where adding $ocase$ hurts performance to a greater degree than subtracting it. According to our results presented so far, we would expect FFNs to be unnecessary for solving the extractive dataset examples, which is possibly why performance is degraded in both cases we intervene, but we don't test this idea in this work.

## C   Effect on Zero-shot Performance

We find that intervening on the model with $\vec{o}$ vectors has applications in controllable generation, that is, guiding the generation process towards some relevant text. We showed this was the case in Section 4, but we can also apply this idea to the context of zero-shot

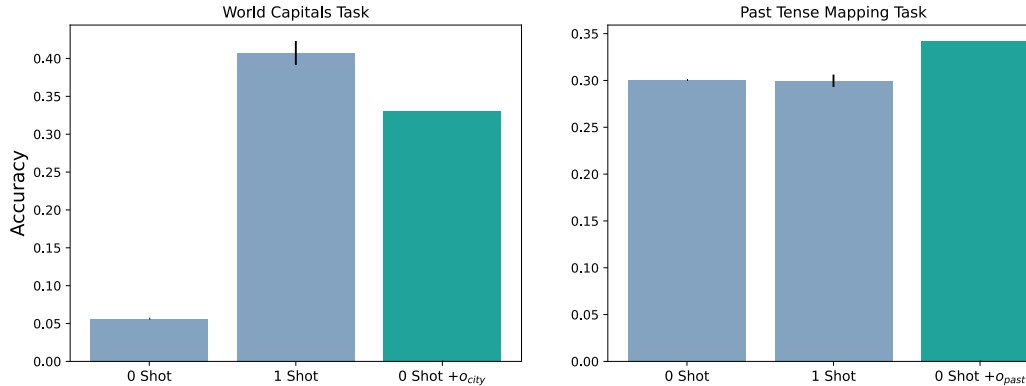

Figure 14: By replacing FFN networks with the corresponding $\vec{o}$ vectors, we show that we can improve zero-shot performance by taking advantage of the model going through argument formation in the zero-shot setting.

learning. When we provide in-context examples, we are also providing the output format of the prompt. Consider the example "Q: What is the capital of Poland? A:". unlike the one shot example given in Figure 2, there is no indication that the next word should be " Warsaw" over continuing the generation as a complete sentence "The capital of Poland is Warsaw", which is what GPT2-Medium actually generates. If we decode at every layer, as is shown in Table C we can see that the model still goes through argument formation despite preferring to generate the full sentence. We can take advantage of this behavior by replacing the FFN layers in the later layers with $\vec{o_{city}}$ in order to guide the generation to the expected response of immediately generating the capital. We can perform this experiment on the past tensing task as well.

| Layer | Top Token |
|---|---|
| 0 | ( |
| 1 | A |
| 2 | A |
| 3 | A |
| 4 | A |
| 5 | A |
| 6 | A |
| 7 | A |
| 8 | A |
| 9 | The |
| 10 | The |
| 11 | The |
| 12 | The |
| 13 | The |
| 14 | The |
| 15 | The |
| 16 | The |
| 17 | The |
| 18 | Poland |
| 19 | Poland |
| 20 | Poland |
| 21 | Poland |
| 22 | Poland |
| 23 | The |

Table 1: These are the top tokens per layer in GPT2-Medium on the example zero-shot Poland example

Results on the zero-shot tasks are shown in Figure 14. We find that on the world capitals task, we can greatly improve the propensity of the model to output the expected answer by performing an $\vec{o}$ vector intervention, improving zero-shot performance from 5.6% to 33.0%. On the past tense mapping task, where perhaps the output format is more obvious from the prompt, the zero and one shot performances are about equal, but we still see a modest improvement over the one shot results of about 4.2%. Although the tasks are very simple, we achieve this by effectively ablating FFN layers (layers 19-23) and precomputing their activations, suggesting it might be possible to edit models extensively to limit their expressiveness to one type of output while also making them more efficient. We are optimistic about future work in this area.

## D  Effect of Layer Choice on Intervention Results

In the main text, we replace FFNs starting at either layer 18 or 19 GPT2-Medium to the end (indexed at 0). We find that intervening on only one layer promotes the output token, but not to the top of the distribution. One possibility is that the model makes gradual updates that are pushing the token representation in generally the same direction (Jastrzebski et al., 2017). In Figure 15, we show that adding any of the $\vec{o}$ vector interventions at any single layer at 18 or afterwards, there is a roughly equivalent increase to the average reciprocal rank of the target word. The logit difference between the argument and answer token (in the logits of each early-decoded layer) shows this as well as a gradual increase. This is exemplified in Figure 2 in the main paper.

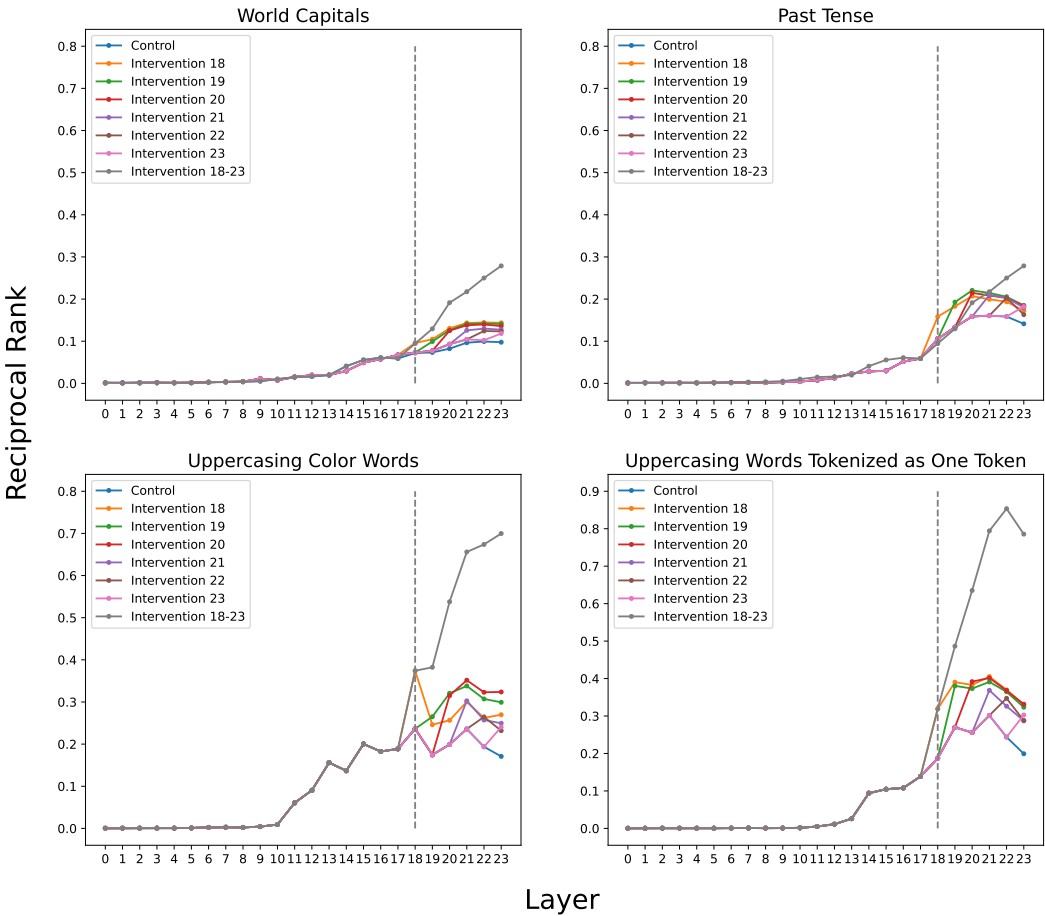

Figure 15: Replacing any individual FFN update is worse than replacing all of them. This supports the idea that networks made gradual updates to their representations, and that the $\vec{o}$ vectors we extract behave this way as well: multiple similar updates are made $k$ layers in a row. Interestingly, the average boost to the reciprocal rank is about the same regardless of which single layer we apply the update at, suggesting that this range of FFNs are operating in same space.

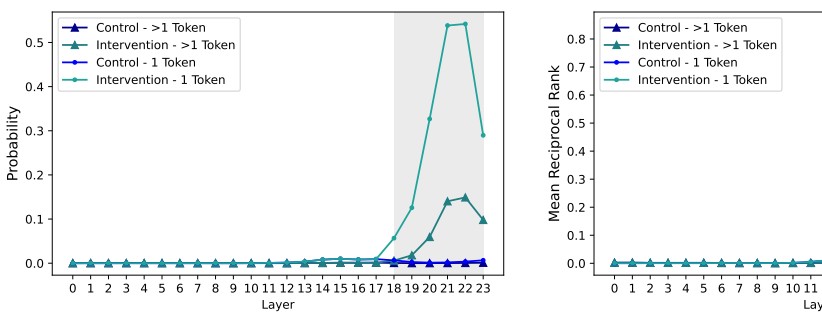

Figure 16: When the uppercase version of a word gets broken down into multiple subtokens, mapping to that token becomes much less probable and is generally harder of an association for the model to make.

## E    Effect of Tokenization on the Effectiveness of $\vec{o}$ Vectors

The tokenizer can split one word into multiple subtokens, such as "Purple" into the tokens "Pur" and "ple". This occurs with words that were less frequent in the training data. We find that this process has a generally negative effect on the performance of the intervention we perform. Intuitively, if we are trying to use $\vec{o_{upper}}$ to capitalize the "purple" token into "Purple", it must map from "purple" (one token) to "Pur". It seems less obvious, then, that the embeddings would encode a linear relationship between these two, since "Pur" is a subtoken in many other words. We explore this specific phenomenon on the random tokens task from Section 4 with the $\vec{o_{upper}}$ intervention. We take 100 single token words that capitalize to a single token, and 100 others that capitalize to words that break down into multiple tokens. Our results can be seen in Figure 16. We find that tokens that get broken up into multiple tokens are less probable than for tokens that capitalize to single token forms.

## F    Additional Tasks: One-to-One, Many-to-One, and Many-to-Many Relations

In the main paper, we show study three one-to-one relations that exhibit the argument/output pattern, but it remains unclear how well this generalizes to other relations. Using six additional tasks, three many-to-X and three new one-to-one, we provide evidence that suggests that the observed mechanism is specific to one-to-one relations, and does not work when mulitple inputs map to one output. This suggests that the model is sensitive to this distinction of relations during pretraining, and the vector arithmetic mechanism structure we observe only presents for the most explicit relations. In Table G, we give examples of the six new tasks, following the same prompt format as the one used in the main paper. In Table G, we break down the relation type of each task and provide the GPT2-Medium accuracy for each one. Figure 17 shows the early decoding patterns for the argument and answer tokens. While the three one-to-one tasks exhibit the initial promotion of the argument token, followed by the answer token on average, the argument token does not become highly promoted on any of the non one-to-one relations.

## G    Compute

All models were run on NVidia RTX 3090s; Bloom was run locally on 3090s in float16 with CPU offloading.

| Task | Example |
|------|---------|
| Animal Hypernyms | ...Q: The anaconda is a kind of what?\nA: (snake/reptile/boa/...) |
| Name to Nationality | ...Q: What is the nationality of Balzac?\nA: (French) |
| Country to Language | ...Q: What is the official language of Argentina?\nA: (Spanish) |
| Adj. to un-Adj. | ...Q: What is the opposite of able?\nA: (unable) |
| 3rd Person Verbs | ...Q: What is the third person singular of become?\nA: (becomes) |
| Noun Plurals | ...Q: What is the plural of album?\nA: albums |

Table 2: Examples from three non-injective and one injective relation. A given animal (anaconda) is a type of snake and reptile, and other snakes/reptiles also exist (many-to-many). Balzac is only French and other people map to French (many-to-one), etc.

| Task | Accuracy (%) | Task Type |
|------|--------------|-----------|
| Animal Hypernyms | 30.4±1.7 | Many-to-Many |
| Name to Nationality | 73.2±2.0 | Many-to-One |
| Country to Language | 71.2±2.4 | Many-to-Many |
| Adj. to un-Adj. | 12.0±1.1 | One-to-One |
| 3rd Person Verbs | 22.4±0.7 | One-to-One |
| Noun Plurals | 51.6±1.7 | One-to-One |

Table 3: One-shot accuracies for each task across 5 random seeds for GPT2-Medium.

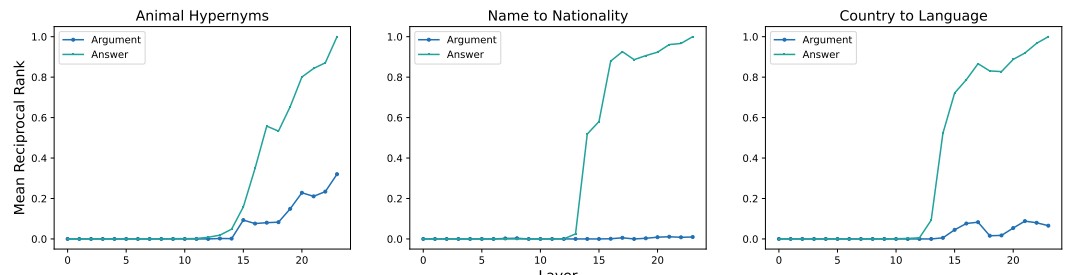

Figure 17: Non-injective tasks show no evidence of argument-function processing on average. In sharp contrast to the past tense, colored objects, capital cities, and un-adj. tasks where this is observed, here, the argument token experiences virtually no spike in reciprocal rank in the intermediate layers.

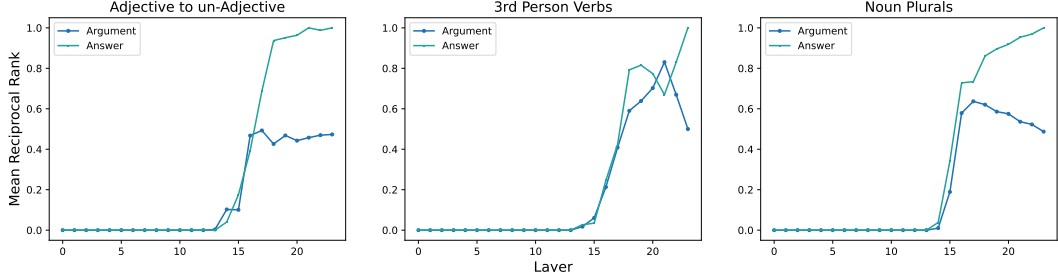

Figure 18: For the first two tasks, the average argument-answer spike pattern is similar to the other one-to-one tasks in which the vector arithmetic analogy held. The results for noun plurals are mostly negative as it appears the model uses argument-function processing only some of the time. We will expand on this in the camera ready paper.

