# OpenReview forum: "A Mechanism for Solving Relational Tasks in Transformer Language Models"
_ICLR.cc/2024/Conference — ICLR 2024 Conference Withdrawn Submission_

### Official Review · Reviewer_76nL · 2023-10-28

**Soundness:** 3 good
**Presentation:** 1 poor
**Contribution:** 2 fair
**Rating:** 3
**Confidence:** 3

**Summary:**

This paper examines how pretrained transformers solve several one-to-one relational tasks using in-context learning and find that a key mechanism is that FFN will apply a simple update on the residual information flow based on the look-up key of the task. This update, when applied in an irrelevant context containing the look-up key, can also increase the probability that the answer is produced. They also show that this mechanism is specific to tasks that require retrieval from pretraining memory, rather than retrieval from the local context, by showing that removing many FFNs may not hurt the performance when the task only requires retrieving from context.

**Strengths:**

* The paper examines how the mechanism found changes with the scale of the model.

* The paper shows a separation between solving tasks using pretrained knowledge or knowledge from retrieval. The result is novel and interesting.

**Weaknesses:**

* The paper's writing requires improvement. The format of the paper is often wrong and really hurts the readability. Also, the legend in Figure 2 is incorrect in the correspondence of color and brings confusion.

* This paper lacks a crucial discussion on some highly relevant previous literature, which may challenge the novelty of the major body of this work. For example, https://arxiv.org/abs/2104.08696 is very well-known and discusses the effect of FFN on relational tasks. To the reviewer's understanding, the main finding of this paper, which is the mechanism that FFN may look up a word/concept and transfer it to some relevant concept, is already shown in this previous work.

**Questions:**

* A key question would be how the authors distinguish themselves with https://arxiv.org/abs/2104.08696. The reviewer also thinks it is crucial to include some other related literature in this field, including the following,

https://arxiv.org/abs/2012.14913

https://arxiv.org/abs/2202.05262

https://arxiv.org/abs/2211.07349

---

### Official Review · Reviewer_P5eV · 2023-10-30

**Soundness:** 2 fair
**Presentation:** 3 good
**Contribution:** 2 fair
**Rating:** 5
**Confidence:** 4

**Summary:**

Here is a summary of the key points from the paper:

Main Contribution:
- The paper presents evidence that large language models (LLMs) like GPT-2 and BLOOM use a simple vector addition mechanism to solve certain relational reasoning tasks in an in-context learning setting.

Novelty:
- The paper offers new insights into the algorithms LLMs use internally to break down complex problems into simple, predictable subprocesses. This contributes to the growing work on "mechanistic interpretability" of LLMs.

Experiments:
- The authors study LLMs ranging from 124M to 176B parameters on tasks involving capital cities, uppercasing, and past-tensing verbs in a few-shot setting.

- They analyze the evolution of next-token predictions during the forward pass and find distinct "stages" - argument formation, function application, and saturation. The stages suggest the model "applies" a function midway through to transform the argument into the answer.

- On GPT-2, they isolate vector updates made by feedforward layers that encode these functions and can induce the function when transferred to unrelated contexts.

- They also show function application relies on factual recall from pretraining, unlike extractive tasks where answers are in context and much of the FFNs can be ablated.

Conclusion:
- Despite the complexity of LLMs, they can exploit simple, intuitive algorithms like vector addition to solve certain relational reasoning tasks. This raises optimism that other complex behaviors may also reduce to understandable subprocesses.

**Strengths:**

- Provides novel insights into the internal mechanisms used by large language models to perform relational reasoning, contributing to the growing work on interpretability.

- Thoroughly analyzes the evolution of predictions during the forward pass across a range of model sizes and layers, revealing staged processing signatures.

- Isolates specific components (FFN update vectors) that implement reusable functions and tests their effects systematically via intervention.

- Connects the identified mechanism to factual recall from pretraining vs utilizing local context, highlighting the specialized roles of different components.

- Experiments comprehensively on multiple datasets testing capital cities, uppercasing, past-tensing, etc. to demonstrate the generalizability of the approach.

- Discusses both the potential of simple underlying mechanisms to explain complex behaviors as well as current limitations and dependencies of the analysis procedures.

- Well written and structured clearly, with figures providing visual summaries of key results.

Overall, the mechanistic analysis is thorough and rigorous, and the identified phenomenon linking staged processing to vector arithmetic for relational reasoning is novel and insightful. The experiments are comprehensive and the paper is very clear.

**Weaknesses:**

- The analysis and intervention procedures require some model-specific customization (e.g. choice of layers for intervention). More work may be needed to generalize the findings.

- directly tested on a couple of models (mainly GPT-2, some GPT-J). More evaluation across diverse model architectures could strengthen claims.

- Mechanism studied is narrow (one-to-one relations). Unclear if this extends to more complex reasoning.

- Lacks direct comparison to other approaches for relational reasoning/in-context learning.

- Some relevant prior work on model interpretability could be discussed more thoroughly.

- Ablation study removes full FFN layers - could also ablate components within FFNs for more precision.

- Intervention experiments introduce the full o vector, leaving some questions about how individual vector components contribute.

- The identified processing stages and arithmetic mechanisms may not be the only or primary way LMs solve these tasks.

Overall the core findings appear quite solid, but expanding the analysis across more models, tasks, and training modes could strengthen the claims. Comparisons to other interpretation methods could better situate the work. Discussion of broader impacts could also be deepened. But these are fairly minor limitations on what is generally a very solid, insightful study.

**Questions:**

1. How well does the identified vector arithmetic mechanism generalize across diverse model architectures, sizes, and training objectives? Is it an inherent inductive bias, or an incidental phenomenon?

2. Do the findings provide any insights into how LMs balance simplicity and effectiveness when selecting internal algorithms? Why this mechanism over others?

3. Can we encourage or discourage this behavior via training/prompting, to improve relational reasoning or reduce undesirable effects?

---

### Official Review · Reviewer_LT5L · 2023-10-31

**Soundness:** 1 poor
**Presentation:** 1 poor
**Contribution:** 2 fair
**Rating:** 1
**Confidence:** 5

**Summary:**

The present paper investigates the internal processes of language models. Specifically, the authors focus on the role of the feed-forward and attention layer to solve relational tasks using acquired knowledge during pre-training and in-context information. To this end, the authors consider three tasks and various pre-trained models. To measure the layers' influence, the authors replace corresponding layers with pre-computed output vectors and conclude that the feed-forward networks within LMs and their attention layers specialize for different roles, with FFNs supporting factual recall and attention copying and pasting from local context.

**Strengths:**

The paper targets the transparency and the understanding of learned mechanisms of black-box models, specifically autoregressive language models. These insights could also be used for pruning or quantization of large models.

**Weaknesses:**

The paper's biggest weakness is its presentation and writing. Therefore, unfortunately, it does not seem to be ready for publication. Starting with Section 2: While 2.1 seems to be an integral part of the methodology, 2.2 and 2.3 seem to describe the experimental setup. A section describing the used notation is completely missing. This makes it hard to follow the paper. I would recommend adding a section on notations and clearly describing the methodology. Instead, Section 3 directly starts with experimental results.

Further issues that make it hard to follow the paper:
- Figure 2: Annotations of colored boxes do not fit with the description in the caption.
- Sec. 2.2 paragraphs are not bold and placed in the middle of the text.
- Non-descriptive section title. Change "1. Intro" to "1. Introduction", 2. <name of the introduced (or investigated) mechanism>
- Vector arrows seem to be misplaced
- Unclear capitalization:
	- "During Function Application"
	- "the model enters Saturation"
	- "GPT2-medium" and "GPT2-Medium"
- Incomplete sentences: For instance, "In particular, the attention layers and most relevant to this work, the FFN modules, which are frequently associated with factual recall and knowledge storage."
- Duplicated bibliography entries and inconsistencies:
	- E.g., Kevin Meng, David Bau, Alex Andonian, and Yonatan Belinkov. Locating and Editing Factual Associations in GPT is referred twice.
	- some provide URLs, some not
	- some provide months, some not
	- the same for the publisher
	- for proceedings sometimes „In Proceedings of …“ sometimes „In <name of conference> …“

- Unclear experiment details and execution:
	- For instance, "1/6th of all FFNs from the top down (e.g., in 24 layer GPT2-Medium: removing the 20-24th FFNs, then the 15-24th etc.)" Removing 1/6 is not corresponding to removing 20-24. It would correspond to removing 21-24.
	- Further, the reproducibility seems to be questionable. A statement in this regard is missing.

**Questions:**

- In Section 4, the authors state that "the target output token becomes the top token in 21.3%, 53.5%, and 7.8% of the time in the last layer in the world capitals, uppercasing, and past tensing tasks, respectively."  What are 21.3, 53.5%, and 7.8% referring to?

---

### Official Review · Reviewer_Kg2J · 2023-11-01

**Soundness:** 2 fair
**Presentation:** 3 good
**Contribution:** 2 fair
**Rating:** 6
**Confidence:** 4

**Summary:**

With empirical evidences on 3 one-to-one abstractive tasks, this paper suggests that LLMs sometimes may employ a simple linear update rule to solve relational tasks. The authors apply LLM's language modeling head $E$ on the residual stream at the last token position of the prompt in different layers and observe that at some middle layer the LM forms a representation of the input/argument (*Poland*) to the function it needs to perform (*capital_of()*). The authors dubb this process as "argument-formation". The later layers "execute" the function on the argument (retrive that the capital of *Poland* is -> *Warsaw*). Connecting with previous works (Meng, Geva) the authors hypothesize that this "execution" (or retrival of information) is done by the FFN layers and they just add a task specific output vector $\vec{o}_{task}$ to the residual stream. This task-specific vector can be obtained with a single stimuli and it can be used to replace FFN computation (specific to the task) in later layers.

**Strengths:**

The problem setting is very interesting, and the paper proposes a surprisingly simple mechanism of how LLMs may solve relational tasks with one-to-one mapping.  This is an interesting insight.

The authors designed an intervention based on their insights and were able to replace some of the later FFN layers contribution by adding a task specific output vector $\vec{o}_{task}$ to the residual stream for each of the layers.

The paper provides empirical evidence of the different role of FFN and attention layers in LLMs which provides some new evidence, complementing and adding insights to recent "dissecting recall" findings of Geva.

**Weaknesses:**

The main weakness is that the empirical evidence is very limited in the sense that it only analyzes three tasks. The results would be strengthened if more tasks could be studied, because that would help clarify whether the phenomenon generalizes (or if not, what the limits might be).  E.g.,  zsRE or counterfact data sets could be sources of larger sets of possible tasks.  It's unclear whether the insights generalize or are very specific to the particular tasks.

The experimental setup is not fully clarified - the paper would be strengthened with a more detailed discussion about how the stimuli (assuming it is not random sampling) and different hyperparameters were selected.  This would aid in reproducibility of the results.

The claim "FFNs can be ablated entirely with relatively minimal performance degradation," is overstated. Some specifics - reciprocal rank is an overly forgiving evaluation metric from which to draw the conclusion. In particular, the intervention often is not strong enough to cause the correct answer to be ranked highest. For example, the target output token becomes the top token in 21.3%, 53.5%, and 7.8% of the time in the last layer in the world capitals, uppercasing, and past-tensing tasks.

The "argument-formation" evidence shown in the paper is very suggestive. But, this paper doesn't provide any insights on why this happens. The paper would be stronger with a more complete discussion, or more experiments explaining:
	* Why does the LM need to do this for some of the tasks? Wouldn't argument information already be present in the earlier token(s) where the argument is actually present as suggested by the Geva 2023 and Meng 2022 work? Can't the LM simply use that information?
	* For the cases where this argument formation doesn't happen cleanly, how does the performance degrade?

**Questions:**

* Will be helpful to clarify: are you calculating $\vec{o}_{task}$ from just a single stimuli? How does the performance change if you use a different example or an alternately structured stimuli?  Would it be possible to improve the quality of that vector measurement by averaging over multiple stimuli?

* Is the selection of the start-end layer the only hyperparam, or does the $\vec{o}_{task}$ vector needs to be scaled up/down as well? (If some scaling is needed, how is that scaling selected, and does that scale factor remain constant across the tasks/layers?)

* Related Work => "Our work builds on the finding that FFN layers promote concepts in the vocabulary space (Geva et al., 2022a) ...". I think Gevas findings were a little bit different, due to the token position of the hidden state examined. They attribute early to mid FFN layers to enrich representation of the last token of the subject. Whereas this paper attributes this role to the later layer FFNs at the last token of the subject.

* A few typos:
	* section 5: "amswer" -> "answer"
	* 5.1: Why *get_capital()* is given as example of an Extractive task? Shouldn't it be an Abstractive one?
	* Appendix:
		* A: "figure 8we"
		* B.1:
			* "As illustrated in the example in Figure ??" -> Missing reference
			* "Sections 2.2 and 2.2" -> ??
			* "$o_case$" or "ocase" -> "$o_{case}$"
		* E: The title is skewed
		* F: Table G -> Table 2
		* Figure 13: Is the x-axis label correct? Should it be "Proportion of FFN Layers **Intact**" instead?